# Tim-3 Coordinates Macrophage-Trophoblast Crosstalk via Angiogenic Growth Factors to Promote Pregnancy Maintenance

**DOI:** 10.3390/ijms24021538

**Published:** 2023-01-12

**Authors:** Liyuan Cui, Fengrun Sun, Yuanyuan Xu, Mengdie Li, Lanting Chen, Chunqin Chen, Jinfeng Qian, Dajin Li, Meirong Du, Songcun Wang

**Affiliations:** Laboratory for Reproductive Immunology, Key Laboratory of Reproduction Regulation (Shanghai Institute of Planned Parenthood Research), Shanghai Key Laboratory of Female Reproductive Endocrine Related Diseases, Hospital of Obstetrics and Gynecology, Fudan University Shanghai Medical College, Shanghai 200031, China

**Keywords:** Tim-3, macrophages, trophoblasts, angiogenic growth factors, pregnancy, placenta

## Abstract

T-cell immunoglobulin mucin-3 (Tim-3) is an important checkpoint that induces maternal–fetal tolerance in pregnancy. Macrophages (Mφs) play essential roles in maintaining maternal–fetal tolerance, remodeling spiral arteries, and regulating trophoblast biological behaviors. In the present study, the formation of the labyrinth zone showed striking defects in pregnant mice treated with Tim-3 neutralizing antibodies. The adoptive transfer of Tim-3^+^Mφs, rather than Tim-3^−^Mφs, reversed the murine placental dysplasia resulting from Mφ depletion. With the higher production of angiogenic growth factors (AGFs, including PDGF-AA, TGF-α, and VEGF), Tim-3^+^dMφs were more beneficial in promoting the invasion and tube formation ability of trophoblasts. The blockade of AGFs in Tim-3^+^Mφs led to the narrowing of the labyrinthine layer of the placenta, compromising maternal–fetal tolerance, and increasing the risk of fetal loss. Meanwhile, the AGFs-treated Tim-3^−^Mφs could resolve the placental dysplasia and fetal loss resulting from Mφ depletion. These findings emphasized the vital roles of Tim-3 in coordinating Mφs-extravillous trophoblasts interaction via AGFs to promote pregnancy maintenance and in extending the role of checkpoint signaling in placental development. The results obtained in our study also firmly demonstrated that careful consideration of reproductive safety should be taken when selecting immune checkpoint and AGF blockade therapies in real-world clinical care.

## 1. Introduction

The establishment of successful placentation and maternal–fetal tolerance are the basis of a successful pregnancy. Extravillous trophoblasts (EVTs) are the dominant cell type involved in the process of placentation, invading deeply into the maternal decidua and uterine blood vessels, dissolving the extracellular matrix, remodeling the uterine vasculature, and coming into direct contact with the maternal decidua immune cells (DICs). The precise regulation of EVT invasion and remodeling of spiral arteries are key events of placentation [1]. As a result of placentation, the maternal immune system has to adapt to tolerate the semi-allogeneic fetus while maintaining maternal immune competence. Inadequate placental development and impaired tolerance induction have been found to be closely related to several pregnancy-associated diseases, including recurrent spontaneous abortion (RSA), pre-eclampsia, and fetal growth retardation [2].

Interactions between fetal-derived EVTs and DICs are critical to placental development. EVTs can modulate DICs to adopt a unique phenotype in order to tolerate the fetus [3,4]. Meanwhile, it has been noticed that DICs present in the vicinity of EVTs and play positive roles in EVT invasion and vascular remodeling [5,6]. As the most important specialized antigen-presenting cells in the decidua, macrophages (Mφs) are the second most abundant leucocytes (comprising 10–20% of the DICs) at the maternal–fetal interface. On the one hand, EVTs are able to modulate Mφ polarization and alter the state of the maternal–fetal immune microenvironment; on the other hand, Mφs can affect the invasiveness and migration of EVTs [7]. Mφs can be polarized into type 1 Mφs (M1) and type 2 Mφs (M2) in the presence of molecular mediators and environmental cues. With high expressions of IL-12, TNF-α, CD80, and CD86, M1 shows greater effectiveness in antigen clearance and switching the T-cell responses to the Th1 immune response. M1 has also been reported to compromise the migration and invasion of trophoblasts [8]. In contrast, with a phenotype with high expressions of IL-10, CD206, CD163, and CD209, M2 has a greater immunosuppressive capacity and plays a constructive role in tissue remodeling, as well as promoting Th2 immune responses. M2 can induce trophoblast migration and invasion [9]. 

T-cell immunoglobulin mucin-3 (Tim-3) is a co-signaling molecule that is widely expressed on the surface of macrophages as well as many other immune cells [10]. The role of Tim-3 in Mφs is complicated and controversial under different microenvironment conditions. For instance, it was reported that Tim-3 inhibits the autoantigen presentation of Mφs by suppressing MHC-II expression, inducing immune tolerance in multiple sclerosis [11]. Meanwhile, in diabetic nephropathy, Tim-3 was also found to promote Mφ activation and aggravate podocyte injury [12]. During pregnancy, Tim-3^+^ decidual Mφs (dMφs) led to the Th2 and Treg bias of decidual CD4^+^T (dCD4^+^T) cells and improved pregnancy maintenance through CD132 [13]. Furthermore, the Tim-3 blockade decreased the phagocytic properties of the dMφs and induced a failure to clear dead and apoptotic cells from the uterus [14].

Our previous study reported that, as a whole, DICs promote EVT function and placental development, and Tim-3 signals regulate this process [6]. We also described the role of Tim-3 in dMφs as a key factor affecting maternal–fetal tolerance in pregnancy. Nonetheless, the interaction between the primary EVTs and specific maternal DIC subsets and the related mechanisms require further study. As dMφs can affect the biological functions of EVTs, can Tim-3 affect placental development by regulating the function of dMφs? If so, what is the mechanism? With these questions in mind, we explored the role of Tim-3 in the crosstalk of dMφs and primary EVTs or HTR8/SVneo cells (immortalized human first-trimester EVTs cell line) in vitro. The involvement of Tim-3/Mφ in the maintenance of pregnancy and development of the placenta was also explored in a normal pregnancy mouse model with Mφ depletion and adoptive transfer treatment.

## 2. Results

### 2.1. Effects of dMφs on the Trophoblasts after Targeting Tim-3

Our study was primed by the discovered effect of the Tim-3 blockade on the labyrinth of pregnant mice. Severe defects in the formation of the labyrinth zone were observed in the pregnant mice that were challenged with Tim-3 blocking antibody (Figure 1A). As dMφs affect the biological behaviors of EVTs and Tim-3 signals regulate dMφ function, we decided to confirm whether the abnormal placental development caused by Tim-3 blockade was a result of dMφ dysfunction. Firstly, we focused on the effects of the dMφs on the trophoblasts after targeting Tim-3. During EVT invasion, two important gelatinases, matrix metalloproteinase-2 (MMP2) and MMP9, are involved in extracellular matrix remodeling [15]. Here, we found that dMφs upregulated the expression of MMP2 and MMP9 in the primary EVTs (Figure 1B). A transwell assay showed an increased number of penetrating HTR8/SVneo cells in co-culture with the dMφs (Figure 1C). As EVTs interact directly with endothelial cells during spiral artery remodeling, we established a co-culture system with human umbilical vein endothelial cells (HUVECs) to assess the tube formation ability of the HTR8/Svneo cells. The dMφs were observed to promote tube formation by the HUVECs and HTR8/SVneo cells co-culture system, implying the ability of the dMφs to promote HUVEC capillary formation (Figure 1D). However, the anti-Tim-3 mAb pretreatment notably attenuated the improvement of the trophoblast invasion and tube formation capacity presented by the dMφs (Figure 1B–D). These observations indicated that, in addition to a direct influence on the immune function of the dMφs themselves [13], Tim-3 blockade also further affected the interaction between the dMφs and EVTs, resulting in placental dysplasia.

### 2.2. Tim-3^+^dMφs Were More Beneficial in Promoting the Invasion and Tube Formation Abilities of Trophoblasts

To further confirm that Tim-3 regulates dMφs function, affecting trophoblast biological behaviors and placental development, RNA-seq was performed to evaluate the differences between the Tim-3^+^ and Tim-3^−^dMφs. The enrichment analysis of the GO term showed that, in addition to immune regulation, the significantly upregulated genes of Tim-3^+^dMφs were mainly enriched in angiogenesis, blood vessel development, sprouting angiogenesis, and so on (Figure 2A). In comparison with the Tim-3^−^dMφs, the Tim-3^+^dMφs showed a stronger capacity in promoting trophoblast invasion (Figure 2B,C) and the tube formation of the HTR8/SVneo cells and HUVECs co-culture system (Figure 2D,E).

The effects of the Mφ depletion and adoptive transfer of the Tim-3^−^Mφs or Tim-3^+^Mφs on the labyrinth zone development were evaluated to provide direct visual insight into the role of Tim-3^+^Mφs in placental development *in vivo*. As shown in Figure 2D, the depletion of Mφs after conception caused a significant narrowing of the placental labyrinth zone. Tim-3^−^Mφs and Tim-3^+^Mφs were isolated from the splenocytes of the pregnant mice and transferred to Mφ-depleted ones, and it was observed that the adoptive transfer of Tim-3^+^Mφs, rather than Tim-3^−^Mφs, could significantly reverse the placental dysplasia resulting from Mφ depletion (Figure 2F).

### 2.3. Tim-3^+^dMφs Promoted the Invasion and Tube Formation Ability of Trophoblasts through Angiogenic Growth Factors

We considered why Tim-3^+^dMφs are more beneficial in promoting the invasion and angiogenesis of trophoblasts than Tim-3^−^dMφs. There were 20 different angiogenesis- and sprouting angiogenesis–related genes between the Tim-3^+^ and Tim-3^−^dMφs (Figure 3A). Among them, *EPAS1* attracted our interest, as the protein product of the *EPAS1* gene is a hypoxia-inducible transcription factor (HIF)-2α. HIF-2α plays a critical role in regulating the cellular functions of trophoblasts [16,17]. Then, we confirmed the difference in the expressions of HIF-2α between the Tim-3^+^ and Tim-3^−^dMφs using flow cytometry (Figure 3B). Furthermore, the frequency of HIF-2α^+^Tim-3^+^Mφs in the RSA patients was much lower than that in normal pregnancy (Figure 3C). However, for unknown reasons, we failed to establish HIF-2α-knockdown cell models in both the primary dMφs and THP1-derived Mφs.

HIF is critical to the upregulation of angiogenic growth factors (AGFs), including vascular endothelial growth factor (VEGF), platelet-derived growth factor (PDGF), transforming growth factor α (TGF-α), angiopoietins, and so on [18,19]. The production of growth factors by the Tim-3^+^dMφs and Tim-3^−^dMφs was detected using a Multi-Analyte Flow Assay Kit. We found that the Tim-3^+^dMφs produced more PDGF-AA, TGF-α, and VEGF than the Tim-3^−^dMφs (Figure 3D). Meanwhile, the production of angiopoietin-2, epidermal growth factor, placental growth factor, and PDGF-BB showed no differences between the Tim-3^+^dMφs and Tim-3^−^dMφs (Appendix A).

Are these AGFs the reason why Tim-3^+^dMφs could promote the invasion and tube formation ability of trophoblasts to a greater extent than the Tim-3^−^dMφs? As shown in Figure 3E–H, the Tim-3^+^dMφs pretreated with anti-PDGF-AA, anti-TGF-α, or anti-VEGF antibody alone slightly attenuated the promoting effect of the Tim-3^+^dMφs on the invasion and tube formation ability of the HTR8/SVneo cells. Additionally, this effect was especially conspicuous in the triple blockade of these AGFs. These observations indicated that Tim-3^+^dMφs promote the invasion and tube formation ability of trophoblasts in an AGF-dependent manner.

### 2.4. The Protective Effect of Tim-3^+^Mφs on Murine Pregnancy Was Counteracted by AGFs Blockade

The adoptive transfer of Tim-3^+^Mφs could markedly alleviate the murine fetal loss [13] and placental dysplasia (Figure 2D) induced by Mφ depletion. We further checked out whether the blockade of AGFs could regulate the effect of Tim-3^+^Mφs adoptive transfer. The blockade of the AGFs counteracted the protective role of Tim-3^+^Mφs adoptive transfer in the pregnancies of Mφ-depleted mice. Compared with the group that underwent transfer with Tim-3^+^Mφs, the additional treatment with anti-PDGF-AA, anti-TGF-α and anti-VEGF antibodies increased the risk of fetal loss (Figure 4A), with a higher rate of embryo resorption (Figure 4B) and narrowed placental labyrinth zone (Figure 4C). The flow cytometry analysis revealed an increased expression of CD80, CD86, and IL-23/23 in the dF4/80^+^ cells from the treated mice (Figure 4D). In addition, the blockade of the AGFs also compromised dCD4^+^T-cell tolerance with the downregulated IL-4 and TGF-β1 expression and upregulated TNF-α and IL-17A expression of the dCD4^+^T cells. In general, these data demonstrated the capacity of Tim-3^+^Mφs for promoting placental development and inducing Th2 and Treg bias in dCD4^+^T cells via AGFs, which is vital for maternal–fetal tolerance and pregnancy maintenance.

### 2.5. AGFs-Treated Tim-3^−^Mφs Enhanced the Invasion and Tube Formation Ability of Trophoblasts and Resolved Murine Fetal Loss Induced by Mφ Depletion

The pretreatment with Tim-3^−^dMφs combined with recombinant PDGF-AA, TGF-α, and VEGF significantly enhanced the invasion of the HTR8/SVneo cells (Figure 5A,B) and the tube formation of the co-culture system of HTR8/SVneo cells and HUVECs (Figure 5C,D). However, the promotion effect of any single AGF treatment was not obvious.

The pretreatment with Tim-3^−^Mφs combined with recombinant PDGF-AA, TGF-α, and VEGF also resolved the fetal resorption and narrowing of the placental labyrinth zone induced by Mφ depletion (Figure 6A–C), accompanied by increased IL-10 expression by the dMφs (Figure 6D) and IL-4 and TGF-β1 expression by the dCD4^+^T cells (Figure 6E). Meanwhile, the CD80, CD86, and IL-23/23 expression by the dMφs (Figure 6D) and TNF-α and IL-17A expression by the dCD4^+^T cells were decreased (Figure 6E). Taken together with our data obtained in vitro and in vivo, Mφs play important roles in the regulation of EVT biological behaviors and maternal–fetal tolerance, and Tim-3 may participate in Mφs-EVTs crosstalk through AGFs and thus play a regulatory role in pregnancy maintenance.

## 3. Discussion

Successful pregnancy is a complicated process involving interactions between the fetal trophoblasts and maternal immune system, which requires maternal tolerance toward the semi-allogeneic fetus as well as sufficient placental formation. It has been recognized that dMφs are stimulated while maintaining the critical equilibrium between functionality and the suppression of excessive inflammation and, ultimately, promoting EVT invasion and placental development [20]. dMφs are recognized as M2 in normal pregnancy. However, an increasing wealth of evidence indicates that the initial classification scheme of dMφs is an over-simplification [21,22]. We previously reported that Tim-3^+^dMφs and Tim-3^−^dMφs were neither precisely M1 nor M2. Compared to Tim-3^−^dMφs, Tim-3^+^dMφs produced more cytokines (both anti-inflammatory and inflammatory) [13]. These results seemed to contrast with the widely accepted notion that the maternal–fetal interface is mainly an anti-inflammatory environment. However, our finding aligned more closely with the hypothesis that immune activation is required to facilitate the invasion of EVTs.

In the current study, we employed primary EVTs from human first-trimester pregnancies and the well-characterized cell line of HTR8/SVneo cells, demonstrating that Tim-3^+^dMφs, rather than Tim-3^−^dMφs, were more effective in promoting the invasion and tube formation ability of trophoblasts. The adoptive transfer of Tim-3^+^Mφs reduced the resorption rate that resulted from Mφ depletion via the improvement of placental development and promotion of the maternal immune responses toward Th2 and Treg bias, further confirming the notable role of the Tim-3^+^dMφs in regulating EVT biological behaviors and maternal–fetal tolerance, leading to the harmonious maternal–fetal crosstalk.

Further investigations showed that the expression of the *EPAS1* gene was higher in Tim-3^+^dMφs than that in Tim-3^−^dMφs. The protein product of the *EPAS1* gene is HIF-2α, which is associated with the activation of Mφs [23], placenta development, and angiogenesis [16,17]. However, for unknown reasons, we failed to establish cell models with HIF-2α-knockdown in both the primary dMφs and THP1-derived Mφs. Although *EPAS1*^+/−^ mice exist, plasmids or lentiviruses that can efficiently knock down HIF-2α in human dMφs require further development.

The differentially expressed HIF-2α inspired us to perform further analyses, as HIF is critical for the upregulation of AGFs [18,19]. We found that the Tim-3^+^dMφs produced more AGFs, including PDGF-AA, TGF-α, and VEGF, than the Tim-3^−^dMφs. These AGFs are reported to take part in the angiogenic process of placentation [24,25]. The promotion effect of the Tim-3^+^dMφs on the invasion and tube formation ability of the HTR8/SVneo cells and the protective role of the Tim-3^+^Mφs in murine pregnancy was notably attenuated by the triple blockade of these AGFs. Meanwhile, the AGF-treated Tim-3^−^Mφs enhanced the invasion and tube formation ability of the trophoblasts and reduced the murine fetal loss that resulted from Mφ depletion. In addition to the pro-angiogenic effect, VEGF can also support the accumulation of Tregs and myeloid cells and facilitate the immunosuppressive activities of Tregs within tumors [26]. TGF-α and PDGF-AA were also reported to partake in the regulation of antitumor immunity [27,28]. The present study confirmed that PDGF-AA, TGF-α, and VEGF affected the tolerance of dMφs and dCD4^+^T cells. Thus, in our study, Tim-3^+^Mφs contributed to pregnancy maintenance via AGFs not only by promoting placental development but also by inducing maternal–fetal tolerance.

As novel strategies for the treatment of chronic infections and tumors, the blockade of co-inhibitory receptors is widely used to improve immune cell responses [29,30]. Tim-3, cytotoxic T-lymphocyte-associated protein 4 (CTLA-4), and programmed cell-death receptor 1 (PD-1) are the major targetable checkpoints of the immune system. The lack of systematic responses has been regarded as the limitation of co-inhibitory receptor blockade in cancer therapies [31]. The crosstalk between the tumor vasculature and immune microenvironment contributes to the immune evasion of tumors; thus, combined therapy regimens targeting both checkpoints and vascular factors can provide a promising strategy for eliciting sustainable and potent antitumor immune responses. For example, injection with combinable antibodies against both PD-1 and TGF-α prolonged the survival of melanoma-bearing mice [28]. In cancer therapy, anti-VEGF treatment enhanced the efficacy of the PD-L1 blockade by regulating either the T-cell responses or Mφ-T-cell crosstalk [32,33]. In pregnancy, anti-Tim-3 clearly suppressed placental development and maternal–fetal tolerance [13]. Although viable pregnancies were reported in patients treated with PD-1 and CTLA-4 checkpoint inhibition [34,35], there was no clinical report regarding the application of Tim-3 antagonists during pregnancy. Furthermore, growth factors derived from DICs are also important for fetal development [36]. To sum up, it is undeniable that the reproductive safety of the pregnancies must be taken into consideration, especially during combined therapy targeting immune checkpoint and/or vascular factors.

In summary (Figure 7), we conclude that Tim-3 signals play regulatory roles in the dialogue between Mφs and trophoblasts in the immune microenvironment at the maternal–fetal interface. EVTs promoted Tim-3^+^Mφs expansion with the engagement of HLA-C/G. With a higher expression of CD132 [13], Tim-3^+^dMφs induced dCD4^+^T cells toward Th2 and Treg bias. Here, Tim-3 was found to not only induce maternal–fetal tolerance but also promote the EVT function through Mφs-EVTs crosstalk dependent on AGFs (including PDGF-AA, TGF-α, and VEGF). These findings contributed to our knowledge of the important mechanism of the Mφ regulation of EVTs during pregnancy, which helps us to understand at least part of the complex processes of human implantation and pregnancy maintenance. The current study also extended our knowledge of the role of checkpoint signaling in placental development. Targeting immune checkpoints and AGFs are regarded as new strategies for antitumor therapy. Again, individualized decisions should be made with careful consideration of the potential benefits and risks for reproductive safety, with an awareness of the risk of adverse pregnancy outcomes and the unknown risk for offspring development.

## 4. Materials and Methods

### 4.1. Human Samples

Decidual and villous tissues of the uterine curettage and whole blood from the peripheral were collected from women who had miscarriages and were diagnosed with RSA, excluding cases of genetic, anatomic, and endocrine abnormalities, infection, etc., (*n* = 8) and women who underwent clinically normal pregnancies of human first-trimester pregnancies that were terminated for non-medical reasons, with no history of spontaneous abortions and at least one successful pregnancy (*n* = 55). Peripheral blood was also collected from normal non-pregnant women of childbearing age (*n* = 9). Primary trophoblast cells were isolated from the freshly collected villi through digestion by trypsin-DNase I (Applichem, Darmstadt, Germany) and discontinuous centrifugation with Percoll gradient, as described previously [37]. DICs were isolated from the decidual tissues, which were subjected to digestion by DNase I (150 U/mL; Applichem, Darmstadt, Germany) and collagenase type IV (1.0 mg/mL, CLS-1; Worthington Biomedical, Lakewood, NJ, USA) in RPMI 1640 (HyClone, Logan, UT, USA) supplement [37]. A magnetic affinity cell-sorting kit (MiltenyiBiotec, North Rhine-Westphalia, Germany) was used for the CD14^+^ cells isolation. The cell sorting of the Tim-3^+^ and Tim-3^−^Mφs was conducted using either the BD FACSAria^TM^ III Cell Sorter with FITC-conjugated anti-human CD14 antibodies and PE-conjugated anti-human Tim-3 antibodies or the Monocyte Isolation Kit with anti-Tim-3 antibodies and Anti-Biotin MicroBeads (MiltenyiBiotec, North Rhine-Westphalia, Germany).

### 4.2. Cell Treatment

The trophoblasts freshly isolated from the villi were seeded in 24-well plates, precoated with diluent Matrigel (Corning, NY, USA) overnight, and cultured at a density of 2 × 10^5^ cells/mL in high-glucose medium (Hyclone, Logan, UT, USA). The HTR8/SVneo cells were cultured with DMEM/F12 medium (Hyclone, Logan, UT, USA) supplemented with 100 U/mL penicillin, 1 μg/mL amphotericin B, 100 μg/mL streptomycin, and 10% fetal bovine serum (FBS) at 37 °C with 5% CO_2_. The sorted Tim-3^+^Mφs, Tim-3^−^Mφs, or dMφs were cultured in RPMI 1640 medium (HyClone, Logan, UT, USA) with 100 U/mL penicillin, 100 μg/mL streptomycin, 1 μg/mL amphotericin B, and 10% FBS at 37 °C with 5% CO_2_. In some cases, the dMφs were pretreated with 10 μg/mL anti-Tim-3 antibodies (clone F38-2E2, Biolegend, San Diego, CA, USA) or 5 μg/mL anti-PDGF-AA (PeproTech, Cranbury, NJ, USA), 5 μg/mL anti-TGF-α (PeproTech, Cranbury, NJ, USA), and 5 μg/mL anti-VEGF (PeproTech, Cranbury, NJ, USA), with all of these three antibodies or isotypes as control. Meanwhile, Tim-3^−^Mφs were treated with PDGF-AA (50 ng/mL; PeproTech, Cranbury, NJ, USA), TGF-α (50 ng/mL; PeproTech, Cranbury, NJ, USA), VEGF (50 ng/mL; PeproTech, Cranbury, NJ, USA), and all three of these growth factors or medium were used as a control. The HUVECs were cultured with DMEM/F12 medium (HyClone, Logan, UT, USA), supplemented with 100 U/mL penicillin, 1 μg/mL amphotericin B, 100 μg/mL streptomycin, and 10% FBS at 37 °C with 5% CO_2_.

### 4.3. Matrigel Invasion Assay

The EVTs or HTR8/SVneo cells (1.6 × 10^4^ in 200 μL of medium) were seeded in the upper chamber (pretreated with Matrigel at 4 °C overnight) in a 24-well plate. In total, 2.5 × 10^5^ Tim-3^+^Mφs, Tim-3^−^Mφs, or dMφs (pretreated with recombinant cytokines or antibodies as described earlier) were seeded with 500 μL medium in the lower chamber and incubated at 37 °C for 48 h. The lower surfaces of the insert chambers were fixed with 4% paraformaldehyde and stained with hematoxylin. After the removal of the Matrigel and non-invading cells on the upper surfaces of the chambers, images were taken with a microscope (BX51tDP70; Olympus, Tokyo, Japan), and the cells were counted under a 100× magnification. The experiments were carried out in triplicate and repeated at least two more times independently.

### 4.4. Tube Formation Assay

The cell tube formation assay was carried out using a 3D Matrigel scaffold in vitro to determine the tube-forming ability of the cells. Pre-cooled Matrigel (9.9 mg/mL, 356234; BD Biosciences, San Jose, CA, USA) at 50 μL per well was carefully coated onto 96-well plates and left for 30 min at 37  °C to polymerize. The HTR8/SVneo cells stained with cell tracker green (C2927; Invitrogen, Waltham, MA, USA), which were pretreated with recombinant growth factors or antibodies, as described earlier, and HUVECs fluorescently stained with cell tracker red (C2925; Invitrogen, Waltham, MA, USA), were both seeded at 2.0 × 10^4^ cells per well with 100 µL DMEM/F12 medium. After 4 h of incubation, the cells were visualized under a microscope, and representative images were taken. The images were later analyzed for junction point counts and tube counts using Image J software (National Institutes of Health, Bethesda, MD, USA). The experiments were carried out in triplicate and repeated at least three times independently.

### 4.5. RNA-Seq Data Analysis

The RNeasy Mini Kit (Qiagen, Hilden, Germany) was used to extract the total RNA of the sorted Tim-3^+^dMφs or Tim-3^−^dMφs, and the purity and integrity of RNA were measured using an Agilent Bioanalyzer 2100 (Agilent Technologies, Santa Clara, CA, USA). The enriched cDNA libraries were sequenced through an Illumina Hi-seq 2500 platform. Sequence alignment between sequenced genes and the human reference genome was performed using STAR. The gene count was acquired using HTSeq-count, and differential expression between Tim-3^+^dMφs and Tim-3^−^dMφs group was analyzed using DE-Seq. The threshold of significant difference was defined as *p* < 0.05. The related signaling pathways enriching differential genes were performed through gene ontology analysis.

### 4.6. Mice

Eight-week-old CBA/J females from Huafukang Bioscience Co. (Beijing, China) were mated with BALB/c males from SLAC Laboratory Animal Co. (Shanghai, China) to induce normal pregnancy (NP). Vaginal plugs inspected in the morning were viewed as indicators of mating, and that morning was regarded as day 0.5 of pregnancy (GD 0.5). Isotype IgG or anti-Tim-3 antibodies (clone RMT3-23, Biolegend, San Diego, CA, USA) i.p. were applied to some pregnant females at dosages of 500, 250, and 250 mg on GD 4.5, GD 6.5, and GD 8.5, respectively, based on our previous publications [13].

For further investigation, Mφ depletion and subsequent adoptive transfer were conducted. Clodronate liposomes were applied to deplete the Mφs at GD 0.5 (200 μL i.p.) and GD 3.5 (100 μL i.p.). Splenocytes of the pregnant CBA/J mice (GD 7.5) were collected and sorted for Tim-3^+^ and Tim-3^−^F4/80^+^ cells using the BD FACSAria^TM^ III Cell Sorter. The sorted cells were resuspended and injected through the tail vein of the Mφ-depleted pregnant mice at GD 4.5. Some of the sorted Tim-3^+^Mφs were treated with anti-PDGF-AA purified (5 μg/mL; Abcam, Waltham, MA, USA), anti-TGF-α purified (5 μg/mL; Santa Cruz, CA, USA), and anti-VEGF (5 μg/mL, clone 2G11-2A05; Biolegend, San Diego, CA, USA). The Tim-3^−^Mφs were stimulated with PDGF-AA (50 ng/mL; PeproTech, Cranbury, NJ, USA), TGF-α (50 ng/mL; PeproTech, Cranbury, NJ, USA), and VEGF (50 ng/mL; PeproTech, Cranbury, NJ, USA) for 48 h *in vitro* before they were used for the transfer. The pregnant mice were monitored on GD 10.5. The embryo absorption rate was calculated as follows: % of resorption = R/(R + V) × 100 (R: the number of hemorrhagic implantations; V: the number of viable, surviving fetuses).

The DICs of the mice were isolated from the freshly collected uteri through digestion by DNase I and collagenase type IV as described previously [13]. For the further flow cytometry of the cytokines, ionomycin (1 μg/mL, Biolegend, San Diego, CA, USA), PMA (50 ng/mL, Biolegend, San Diego, CA, USA), and brefeldin A (10 mg/mL, BioLegend, San Diego, CA, USA) were added 4 h before the intracellular analysis of the T cells.

### 4.7. Hematoxylin and Eosin (H&E) Staining of Placental Hemisections

Paraffin-embedded placentae were sagittally cut into 3 µm thick sections after being fixed in 4% paraformaldehyde overnight at 4 °C. The major sections near the middle of the placentae were prepared for H&E staining. Labyrinth vascular regions were observed using a fluorescence microscope (Olympus, Tokyo, Japan). An average of labyrinth vascular regions of each placenta was measured according to labyrinth vascular regions of five sections at an interval of at least 40 µm.

### 4.8. Flow Cytometry

Flow cytometry was applied to analyze the cell surface molecule expression and intracellular cytokine expression. The antibodies used were as follows: FITC-conjugatedanti-mouse CD4; F4/80; TNF-α; IFN-γ; anti-human CD14; eFluor^®^ 488-conjugated anti-mouse TNF-α; PE-conjugated anti-mouse CD8; Tim-3; TGF-β1; IL-10; anti-human Tim-3; PerCP/Cy5.5-conjugated anti-mouse IL-17A; PE/CY7-conjugated anti-mouse F4/80; IL-10; TNF-α; IL-12/23; APC-conjugated anti-mouse F4/80; Tim-3; TNF-α; IL-10; Brilliant Violet 421-conjugated anti-mouse TGF-β1; CD206; TNF-α; IL-4; Brilliant Violet 510-conjugated anti-mouse CD86; TNF-α; CD4; Brilliant Violet 605-conjugated anti-mouse IL-17A; CD4 (Biolegend, San Diego, CA, USA). The HIF-2α antibody (Santa Cruz, CA, USA) was pretreated with the Lightning-Link APC Conjugation Kit (Innova Biosciences, Cambridge, UK). Before intracellular staining, a Fix/Perm kit (Biolegend, San Diego, CA, USA) was used to fix and permeabilize cells. Flow cytometry was carried out using a Beckman-Coulter CyAn ADP cytometer (Beckman-Coulter, Bria, CA, USA) and analyzed using FlowJo software (Tree Star, Ashland, OR, USA).

The production of Angiopoietin-2, EGF, FGF, TGF-α, PDGF-AA, PDGF-BB, and VEGF by EVTs was evaluated using a Multi-Analyte Flow Assay Kit (Human Growth Factor Panel, Biolegend, San Diego, CA, USA).

### 4.9. Statistical Analysis

Throughout the study, all the variables were distributed normally. The variables were presented as means with standard errors. Statistical analyses were conducted using GraphPad Prism 5 or 8 software (GraphPad, San Diego, CA, USA). One-way analysis of variance (ANOVA) was applicated to evaluate the differences. In addition, *p* < 0.05 were considered statistically significant. In cases where *p* < 0.05 in ANOVA, we also performed the post-hoc Dunnett *t*-test to determine the differences between each group.

## Figures and Tables

**Figure 1 ijms-24-01538-f001:**
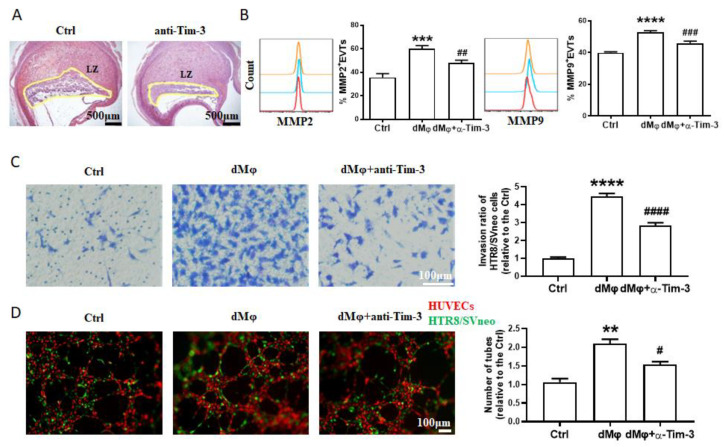
Effects of dMφs on trophoblasts after targeting Tim-3. (**A**) Representative placental hemisections with H&E staining from GD 10.5 mice. Tim-3 blockade caused significantly narrowed placental labyrinth zone (LZ), *n* = 3–6 mice per group. (**B**) MMP2 and MMP9 by extravillous trophoblasts (EVTs) after treatment with dMφs (pretreated with or without 10 μg/mL anti-Tim-3). (**C**) Matrigel invasion assays after indicated treatment of HTR8/SVneo cells. (**D**) Immunofluorescent assay for the 3D tubes formed by HTR8/SVneo cells (green) and HUVECs (red) following treatment with dMφs (pretreated with or without 10 μg/mL anti-Tim-3) at 4 h. Data represent the mean ± standard error of the mean (SEM). ** *p* < 0.01, *** *p* < 0.001, **** *p* < 0.0001, compared with the control group. # *p* < 0.05, ## *p* < 0.01, ### *p* < 0.001, #### *p* < 0.0001, compared with the group of dMφ.

**Figure 2 ijms-24-01538-f002:**
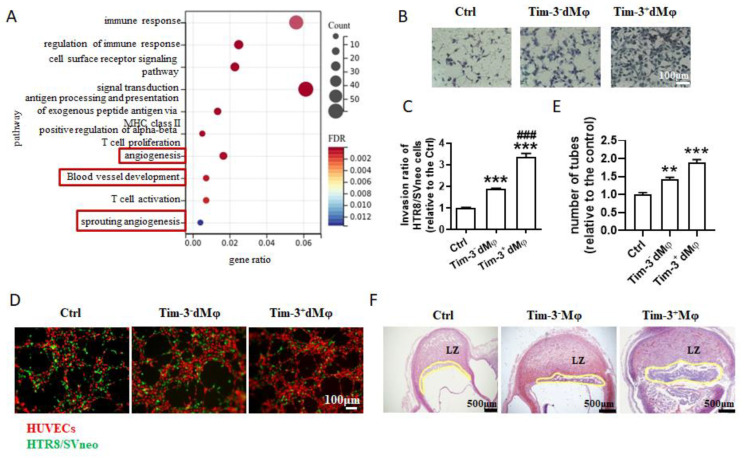
Tim-3^+^dMφs were more beneficial to promote invasion and tube formation ability of trophoblasts. (**A**) Enriched Gene Ontology term pathways of differential gene clustering between Tim-3^−^dMφs and Tim-3^+^Mφs from human normal first-trimester pregnancy. (**B**,**C**) Matrigel invasion assays of HTR8/Svneo cells following treatment of Tim-3^−^dMφs or Tim-3^+^dMφs. Data represent the mean ± SEM. ** *p* < 0.01, *** *p* < 0.001, compared with the ctrl group. ### *p* < 0.001, compared with the group of Tim-3^−^dMφ. (**D**,**E**) Immunofluorescent assay at 4 h of the 3D tube formation by HTR8/SVneo cells (green) and HUVECs (red) with indicated treatments. (**F**) H&E-stained placental hemisections from pregnant mice with Mφ depletion and adoptive transfer of Tim-3^+^Mφs or Tim-3^−^Mφs at GD 10.5, *n* = 3–6 mice per group.

**Figure 3 ijms-24-01538-f003:**
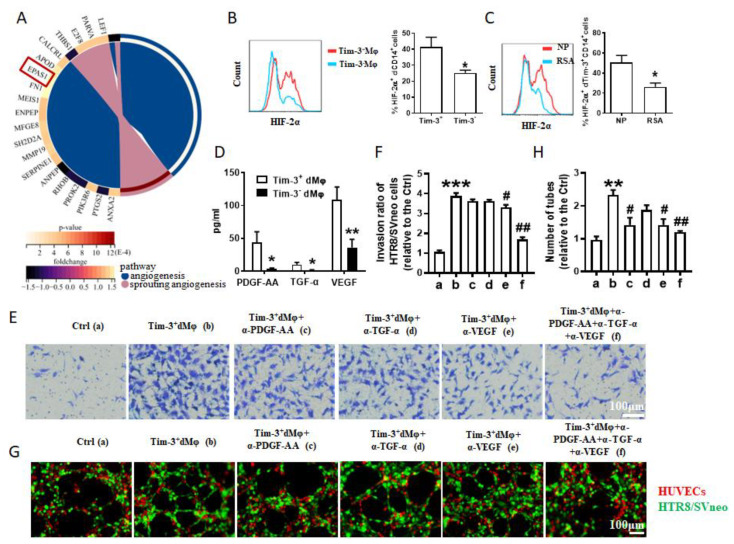
Tim-3^+^dMφs promoted invasion and tube formation ability of trophoblasts through angiogenic growth factors. (**A**) Differential expression of angiogenesis- and sprouting angiogenesis–related mRNAs between Tim-3^+^dMφs and Tim-3^−^dMφs from RNA microarray assay. (**B**) Flow cytometry plots and quantifications of HIF-2α (the protein product of *EPAS1* gene) by human Tim-3^+^dMφs and Tim-3^−^dMφs (*n* = 9). * *p* < 0.05. (**C**) Flow cytometric analysis and quantification of HIF-2α on human Tim-3^+^dMφs between normal pregnancies (NP, *n* = 11) and patients diagnosed with recurrent spontaneous abortion (RSA, *n* = 8). * *p* < 0.05. (**D**) Levels of PDGF-AA, TGF-α, and VEGF in supernatants of Tim-3^+^ and Tim-3^−^dMφs. * *p* < 0.05. (**E**,**F**) Matrigel invasion assays of HTR8/SVneo cells following treatment with Tim-3^+^dMφs (pretreated with or without anti-PDGF-AA antibody, and/or anti-TGF-α antibody, and/or anti-VEGF antibody). (**G**,**H**) Immunofluorescent assay at 4 h of the 3D tube formation by HTR8/SVneo cells (green) and HUVECs (red) with indicated treatment. Data represent the mean ± SEM. ** *p* < 0.01, *** *p* < 0.001, compared with group a. # *p* < 0.05, ## *p* < 0.01, compared with group b. a: Ctrl, b: Tim-3^+^dMφs, c: Tim-3^+^dMφs+α-PDGF-AA, d: Tim-3^+^dMφs+α-TGF-α, e: Tim-3^+^dMφs+α-VEGF, f: Tim-3^+^dMφs+α-PDGF-AA+α-TGF-α+α-VEGF.

**Figure 4 ijms-24-01538-f004:**
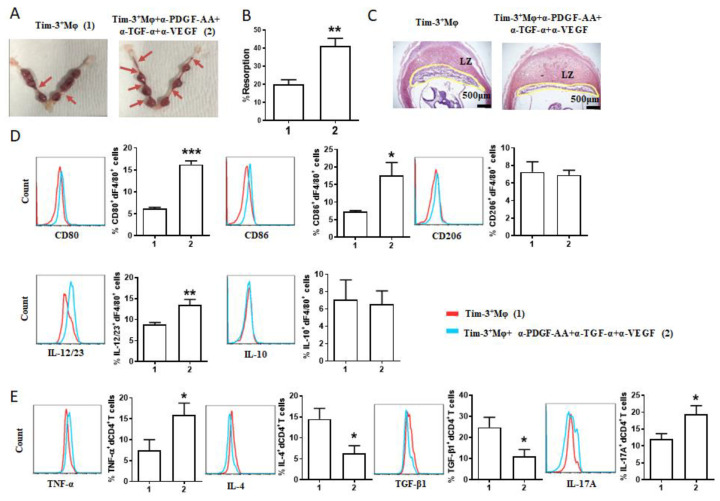
The protective effect of Tim-3^+^Mφs on murine pregnancy was counteracted by AGF blockade. (**A**–**C**) Representative images of uterus (**A**), statistics of the percentage of fetal resorption (**B**), and representative images of placental hemisections (H&E-stained) (**C**) of pregnant mice after Mφ depletion and adoptive transfer of indicated Tim-3^+^Mφs. Fetal loss sites were identified as necrosis and hemorrhagic spots (shown with red arrows). (**D**) Representative images of flow cytometry and quantifications of surface molecule and cytokine expression of dF4/80^+^ cells from pregnant mice after Mφ depletion and the indicated adoptive transfer. (**E**) Cytokine expression of dCD4^+^ T cells of pregnant mice after Mφ depletion and the indicated adoptive transfer. Data represent mean ± SEM (*n* = 5–11). * *p* < 0.05, ** *p* < 0.01, *** *p* < 0.001, compared with group 1.

**Figure 5 ijms-24-01538-f005:**
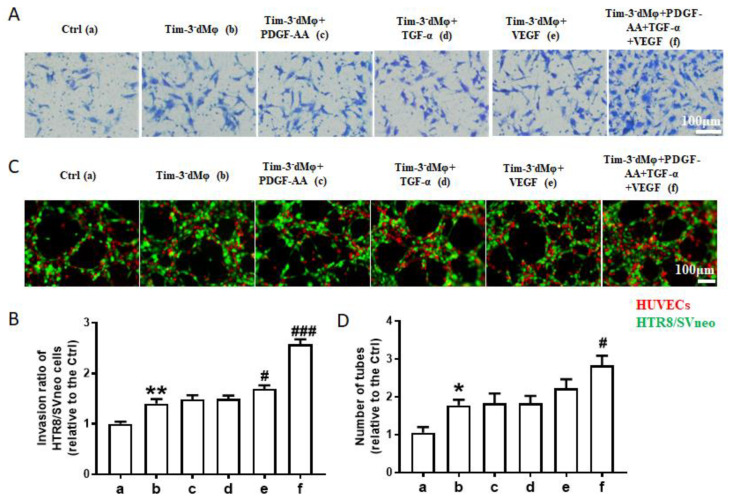
**AGFs-treated Tim-3^−^Mφs enhanced the invasion and tube formation ability of trophoblasts**. (**A**,**B**) Matrigel invasion assays of HTR8/SVneo cells treated with indicated Tim-3^−^dMφs (pretreated with or without PDGF-AA, and/or TGF-α, and/or VEGF). (**C**,**D**) Immunofluorescent assay at 4 h of the 3D tube formation by HTR8/SVneo cells (green) and HUVECs (red) with indicated treatment. Data represent the mean ± SEM. * *p* < 0.05, ** *p* < 0.01, compared with the group a. # *p* < 0.05, ### *p* < 0.001, compared with group b. a: Ctrl, b: Tim-3^−^dMφs, c: Tim-3^−^dMφs+PDGF-AA, d: Tim-3^−^dMφs+TGF-α, e: Tim-3^−^dMφs+VEGF, f: Tim-3^−^dMφs+PDGF-AA+TGF-α+VEGF.

**Figure 6 ijms-24-01538-f006:**
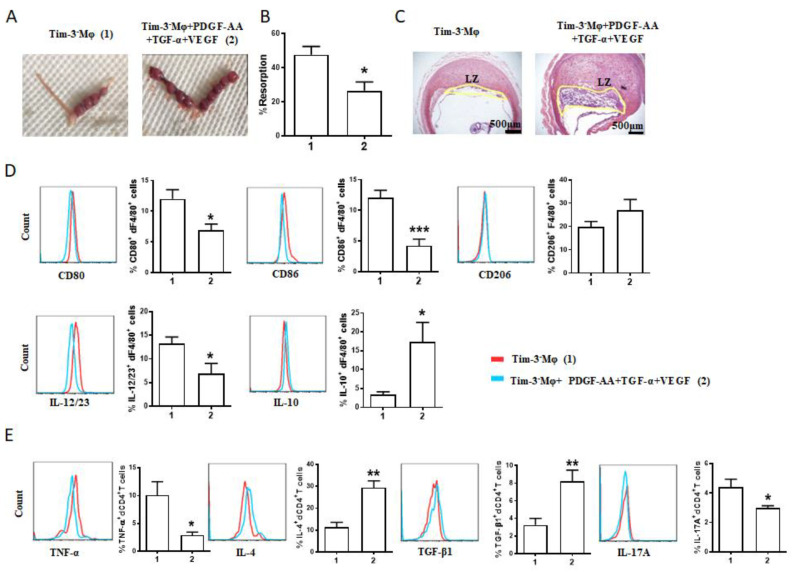
AGFs-treated Tim-3^−^Mφs also alleviated murine fetal loss resulting from Mφ depletion. (**A**–**C**) Representative images of uterus (**A**), statistics of the percentage of fetal resorption (**B**), and placental hemisections (H&E-stained) (**C**) of pregnant mice after Mφ depletion and adoptive transfer of indicated Tim-3^−^Mφs. (**D**) Representative images of flow cytometry and quantifications of surface molecule and cytokine expression on dMφs from pregnant mice after Mφ depletion and adoptive transfer. (**E**) Cytokine expression of dCD4^+^ T cells from pregnant mice after Mφ depletion and the indicated adoptive transfer. Data represent mean ± SEM (*n* = 4–8). * *p* < 0.05, ** *p* < 0.01, *** *p* < 0.001, compared with group 1.

**Figure 7 ijms-24-01538-f007:**
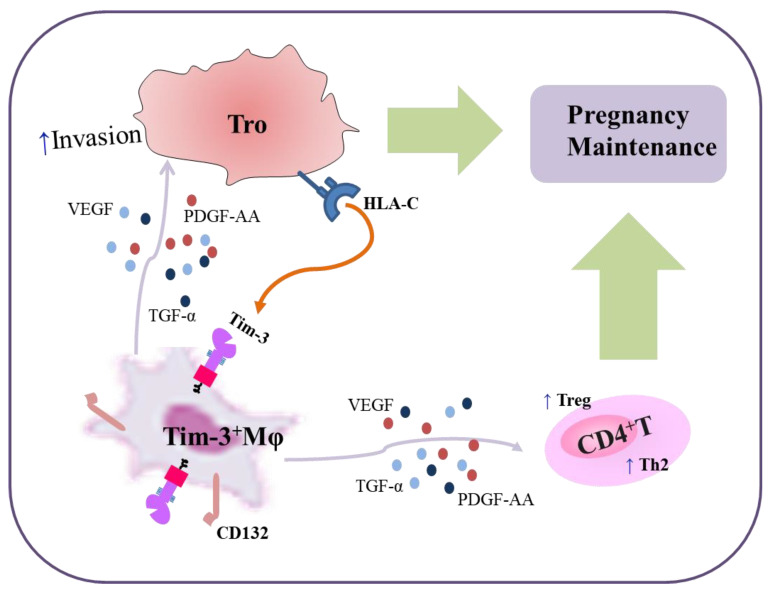
Schematic diagram of functional regulation of Tim-3 on Mφs-trophoblasts crosstalk during pregnancy. Previously, we reported EVTs induced the higher Tim-3 expression on dMφs in normal pregnancy in an HLA-C-dependent manner. With higher expression of CD132, Tim-3^+^dMφs induced dCD4^+^T cells toward Th2 and Treg bias and promoted pregnancy maintenance. The current study revealed that Tim-3^+^dMφs promoted invasion and tube formation ability of trophoblasts through AGFs (PDGF-AA, TGF-α, and VEGF). These AGFs derived from Tim-3^+^dMφs also helped to promote Th2 and Treg bias in dCD4^+^T cells. The blockade of Tim-3 or AGFs resulted in placental dysplasia and dysfunction of maternal–fetal tolerance. While AGFs-treated Tim-3^−^Mφs could also partially resolve the fetal resorption resulting from Mφ depletion. Thus, Tim-3 not only promoted maternal–fetal tolerance but also improved EVT function through Mφs-EVTs interaction to promote pregnancy maintenance.

## Data Availability

The original data are available upon reasonable request. The data are not publicly available due to the data will be used in subsequent studies.

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
