# Peer review of "Tim-3 Coordinates Macrophage-Trophoblast Crosstalk via Angiogenic Growth Factors to Promote Pregnancy Maintenance"

_ijms, 2023, doi:10.3390/ijms24021538_

Round 1

Reviewer 1 Report

Dear authors, I have read with great interest the manuscript titled: "Tim-3 coordinates macrophage-trophoblast crosstalk via angiogenic growth factors to promote pregnancy maintenance".

This study is really interesting. The work is enough attractive for the readers, well written and well prepared.  Figures and Tables are complete and adequate. Materials and methods are clearly explained. The discussion supports the results of the research. The synthesis of the work is excellent to understand the study objective and to understand the findings of the study.

Author Response

Reviewer 1:

Dear authors, I have read with great interest the manuscript titled: "Tim-3 coordinates macrophage-trophoblast crosstalk via angiogenic growth factors to promote pregnancy maintenance".

This study is really interesting. The work is enough attractive for the readers, well written and well prepared.  Figures and Tables are complete and adequate. Materials and methods are clearly explained. The discussion supports the results of the research. The synthesis of the work is excellent to understand the study objective and to understand the findings of the study.

Response:Thank you so much for your kind comments. 

Reviewer 2 Report

Tim-3 coordinates macrophage-trophoblast crosstalk via angiogenic growth factors to promote pregnancy maintenance, an original work by Cui et al. reveal that With higher production of angiogenic growth factors (AGFs, including PDGF-AA, TGF-α and VEGF), Tim-3 +dMφs were more beneficial to promote invasion and tube formation ability of trophoblasts. Adoptive transfer of Tim-3 +Mφs, rather than Tim-3 -Mφs, reversed murine placental dysplasia resulted from Mφ depletion. Blockade of AGFs in Tim-3 +Mφs leaded to the narrowing of labyrinthine layer of placenta, compromising of maternal-fetal tolerance and increasing risk of fetal loss. While AGFs treated Tim-3 -Mφs could rescue the placental dysplasia and fetal resorption resulted from Mφ depletion. The finding is interesting, however the abstract section should be improved. I provided some comments on the manuscript. Concerns should be addressed by authors.

1.     English grammar and typo errors must to be corrected.

2.     Rephrase and better the abstract.

3.     Increase quality of Figure 2 and 4.

4.     What is the role of CD132? 

Author Response

Reviewer 2:

Tim-3 coordinates macrophage-trophoblast crosstalk via angiogenic growth factors to promote pregnancy maintenance, an original work by Cui et al. reveal that With higher production of angiogenic growth factors (AGFs, including PDGF-AA, TGF-α and VEGF), Tim-3 +dMφs were more beneficial to promote invasion and tube formation ability of trophoblasts. Adoptive transfer of Tim-3 +Mφs, rather than Tim-3 -Mφs, reversed murine placental dysplasia resulted from Mφ depletion. Blockade of AGFs in Tim-3 +Mφs leaded to the narrowing of labyrinthine layer of placenta, compromising of maternal-fetal tolerance and increasing risk of fetal loss. While AGFs treated Tim-3 -Mφs could rescue the placental dysplasia and fetal resorption resulted from Mφ depletion. The finding is interesting, however the abstract section should be improved. I provided some comments on the manuscript. Concerns should be addressed by authors.

  1. English grammar and typo errors must to be corrected.

Response: Thank you so much for your helpful comments. We have asked for language editing from Multidisciplinary Digital Publishing Institute (MDPI), which is a professional English editing company, located in Basel, Switzerland. Hope the language is much better now. We also attached the certificate of English editing of MDPI.

  1. Rephrase and better the abstract.

Response: Thank you so much for your helpful comments. We rewrote the abstract, hope it is much better now.

  1. Increase quality of Figure 2 and 4.

Response: Thank you so much for your helpful comments. We increased the quality of Figure 2 and 4, hope the quality is much better now.

  1. What is the role of CD132? 

Response: Thank you so much for your kind comments. We conducted microarray profiling on highly purified populations isolated by FCM to understand what genomic differences distinguish Tim-3+Mφs and Tim-3-Mφs subsets. Further analysis revealed that IL2RG gene were overexpressed in the Tim-3+dMφs population as compared with the Tim-3-dMφs. The protein product of IL2RG gene is CD132, also known as the common gamma chain (a type I cytokine receptor). CD132-dependent cytokines, including IL-2, IL-4, IL-7, IL-9, IL-15 and IL-21, play crucial roles in the proliferation, survival and differentiation of multiple cell lineages of both the innate and adaptive immune systems (Pulliam SR, et al. Immuno Lett 2016). Tim-3+dMφs and Tim-3-dMφs showed CD132 expressional differences resulting in unique function on regulating decidual CD4+T cell tolerance. With higher CD132 expression, Tim-3+dMφs subset induced Th2 and Treg bias in decidual CD4+T cells and promoted pregnancy maintenance. Blockade of Tim-3 or CD132 pathways leaded to the dysfunction of maternal-fetal tolerance and increased fetal loss. These results have been published on the magazine of Cell Death Dis (MD Li, et al. Cell Death Dis 2022).

Reviewer 3 Report

Tim-3 coordinates macrophage-trophoblast crosstalk via angiogenic growth factors to promote pregnancy maintenance.

Note:

I had no access to supplementary figures, nor did the manuscript has line numbers to allow for easy reference in my notes.

The paper is interesting, and much effort was put into the work presented. I have a few minor points listed below.

Minor Revisions:

1.      I suggest adding the word “placenta” to the list of keywords.

2.      Review for the use of proper scientific naming of genes/proteins (human vs mice), minor typos, formatting, punctuation, and frequent grammar problems throughout the manuscript/figures. Grammar/ word usage issues made it difficult to read and, sometimes, to accurately understand the authors’ intended meaning.

3.      On page 7, “However, we failed to establish HIF-2α-knockdown cell models in both primary dMφs and THP1-derived Mφs for unknown reasons.” and, again, in the discussion section; can you add speculation of why that happened based on what we know of HIF2A function?

4.      On page 8, the authors wrote, “In general, these data proved the Tim-3+Mφs capacity of promoting placental development and inducing Th2 and Treg bias in dCD4+T cells via AGFs, which is important for maternal-fetal tolerance and pregnancy maintenance.” While this sentence requires some grammar checking, I also recommend not using the word “proved,” instead using something like “indicate” or “demonstrate.”

5.      There is no need to reference figures in the discussion.

Figure Revisions:

Figure 1:

1.      Insert a scale bar on the images and, if possible, increase the resolution.

2.      In the figure legend, it was mentioned that the values represent mean ± SEM but in the materials and methods, it was mentioned that the data represent mean ± SD, so which one is correct?

3.      Fig. 1D, is the number of tubes that of both HTR8/SVneo and HUVEC, or each cell type was quantified separately?

Figure 2:

1.      Insert a scale bar on the images and if possible, increase the resolution.

2.      In the figure legend, it was mentioned that the values represent mean ± SEM but in the materials and methods, it was mentioned that the data represent mean ± SD, so which one is correct?

3.      Fig 2B: why the invasion quantification graph uses a different metric/unit than that used in the rest of the figures? 

4.      Fig 2C: please add a quantification graph for the tube formation images.

Figure 3:

1.      Insert a scale bar on the images and, if possible, increase the resolution.

2.      In the figure legend, it was mentioned that the values represent mean ± SEM but in the materials and methods, it was mentioned that the data represent mean ± SD, so which one is correct?

3.      Fig. 3H, is the number of tubes that of both HTR8/SVneo and HUVEC, or each cell type was quantified separately?

Figure 4:

1.      Insert a scale bar on the images and, if possible, increase the resolution of all panels.

2.      In the figure legend, it was mentioned that the values represent mean ± SEM but in the materials and methods, it was mentioned that the data represent mean ± SD, so which one is correct?

Figure 5:

1.      Insert a scale bar on the images and, if possible, increase the resolution.

2.      In the figure legend, it was mentioned that the values represent mean ± SEM but in the materials and methods, it was mentioned that the data represent mean ± SD, so which one is correct?

3.      Fig. 5D, is the number of tubes that of both HTR8/SVneo and HUVEC, or each cell type was quantified separately?

Figure 6:

1.      Insert a scale bar on the images and increase the resolution.

2.      In the figure legend, it was mentioned that the values represent mean ± SEM but in the materials and methods, it was mentioned that the data represent mean ± SD, so which one is correct?

Figure 7:

1.      I think it is better if the arrow after the words’ invasion, Treg, and Th2 is moved before the words rather than after (so it gives the meaning of “promote/increase invasion”).

Author Response

Reviewer 3:

I had no access to supplementary figures, nor did the manuscript has line numbers to allow for easy reference in my notes.

The paper is interesting, and much effort was put into the work presented. I have a few minor points listed below.

Response: Thank you so much for your kind comments. We added line numbers and supplementary figure in the revised manuscript.  

Minor Revisions:

  1. I suggest adding the word “placenta” to the list of keywords.

Response: Thank you so much for your helpful comments. We added “placenta” to the list of keywords.

  1. Review for the use of proper scientific naming of genes/proteins (human vs mice), minor typos, formatting, punctuation, and frequent grammar problems throughout the manuscript/figures. Grammar/ word usage issues made it difficult to read and, sometimes, to accurately understand the authors’ intended meaning.

Response: Thank you so much for your helpful comments. We have asked for language editing from Multidisciplinary Digital Publishing Institute (MDPI), which is a professional English editing company, located in Basel, Switzerland. Hope the language is much better now. We also attached the certificate of language editing by MDPI.

  1. On page 7, “However, we failed to establish HIF-2α-knockdown cell models in both primary dMφs and THP1-derived Mφs for unknown reasons.” and, again, in the discussion section; can you add speculation of why that happened based on what we know of HIF2A function?

Response: Thank you so much for your kind comments. Since HIF-2α whole-body knockout mice are embryonic lethal (Tian H, et al. Genes Dev 1998), it is likely that the macrophages with this gene knockout die, so the live cell test showed that this gene was not knocked out. Though EPAS1+/− mice exists, plasmids or lentiviruses that can efficiently knock down HIF-2α in human dMφs need further development.

  1. On page 8, the authors wrote, “In general, these data proved the Tim-3+Mφs capacity of promoting placental development and inducing Th2 and Treg bias in dCD4+T cells via AGFs, which is important for maternal-fetal tolerance and pregnancy maintenance.” While this sentence requires some grammar checking, I also recommend not using the word “proved,” instead using something like “indicate” or “demonstrate.”

Response: Thank you so much for your helpful comments. We used “demonstrated” instead of “proved” in the revised manuscript.

  1. There is no need to reference figures in the discussion.

 Response: Thank you so much for your helpful comments. We deleted “Figure 1A” in the revised manuscript.

Figure Revisions:

Figure 1:

  1. Insert a scale bar on the images and, if possible, increase the resolution.

 Response: Thank you so much for your helpful comments. We added a scale bar on the images and increased the resolution, hope the figure is much better now.

  1. In the figure legend, it was mentioned that the values represent mean ± SEM but in the materials and methods, it was mentioned that the data represent mean ± SD, so which one is correct?

 Response: Sorry for making you confused. It should be standard error of the mean (SEM). We changed it in the revised manuscript.

  1. 1D, is the number of tubes that of both HTR8/SVneo and HUVEC, or each cell type was quantified separately?

Response: Thank you so much for your helpful comments. The number of tubes is that of both HTR8/SVneo and HUVEC.

Figure 2:

  1. Insert a scale bar on the images and if possible, increase the resolution.

 Response: Thank you so much for your helpful comments. We added a scale bar on the images and increased the resolution, hope the figure is much better now.

  1. In the figure legend, it was mentioned that the values represent mean ± SEM but in the materials and methods, it was mentioned that the data represent mean ± SD, so which one is correct?

Response: Sorry for making you confused. It should be standard error of the mean (SEM). We changed it in the revised manuscript.

  1. Fig 2B: why the invasion quantification graph uses a different metric/unit than that used in the rest of the figures? 

 Response: Sorry for making you confused. We unified the metric/unit of the invasion quantification graphs in the revised manuscript.

  1. Fig 2C: please add a quantification graph for the tube formation images.

 Response: Thank you so much for your helpful comments. We added a quantification graph for the tube formation images (Figure 2E) 

Figure 3:

  1. Insert a scale bar on the images and, if possible, increase the resolution.

 Response: Thank you so much for your helpful comments. We added a scale bar on the images and increased the resolution, hope the figure is much better now.

  1. In the figure legend, it was mentioned that the values represent mean ± SEM but in the materials and methods, it was mentioned that the data represent mean ± SD, so which one is correct?

Response: Sorry for making you confused. It should be standard error of the mean (SEM). We changed it in the revised manuscript.

  1. 3H, is the number of tubes that of both HTR8/SVneo and HUVEC, or each cell type was quantified separately?

Response: Thank you so much for your helpful comments. The number of tubes is that of both HTR8/SVneo and HUVEC cells.

Figure 4:

  1. Insert a scale bar on the images and, if possible, increase the resolution of all panels.

Response: Thank you so much for your helpful comments. We added a scale bar on the images and increased the resolution, hope the figure is much better now.

  1. In the figure legend, it was mentioned that the values represent mean ± SEM but in the materials and methods, it was mentioned that the data represent mean ± SD, so which one is correct?

Response: Sorry for making you confused. It should be standard error of the mean (SEM). We changed it in the revised manuscript.

Figure 5:

  1. Insert a scale bar on the images and, if possible, increase the resolution.

Response: Thank you so much for your helpful comments. We added a scale bar on the images and increased the resolution, hope the figure is much better now.

  1. In the figure legend, it was mentioned that the values represent mean ± SEM but in the materials and methods, it was mentioned that the data represent mean ± SD, so which one is correct?

Response: Sorry for making you confused. It should be standard error of the mean (SEM). We changed it in the revised manuscript.

  1. 5D, is the number of tubes that of both HTR8/SVneo and HUVEC, or each cell type was quantified separately?

Response: Thank you so much for your helpful comments. The number of tubes is that of both HTR8/SVneo and HUVEC cells.

Figure 6:

  1. Insert a scale bar on the images and increase the resolution.

Response: Thank you so much for your helpful comments. We added a scale bar on the images and increased the resolution, hope the figure is much better now.

  1. In the figure legend, it was mentioned that the values represent mean ± SEM but in the materials and methods, it was mentioned that the data represent mean ± SD, so which one is correct?

Response: Sorry for making you confused. It should be standard error of the mean (SEM). We changed it in the revised manuscript.

Figure 7:

  1. I think it is better if the arrow after the words’ invasion, Treg, and Th2 is moved before the words rather than after (so it gives the meaning of “promote/increase invasion”).

Response: Thank you so much for your helpful comments. We changed them in the revised manuscript.
